# Minibatch Stochastic Approximate Proximal Point Methods

**Hilal Asi**[*]
Stanford University
asi@stanford.edu

**Karan Chadha**[*]
Stanford University
knchadha@stanford.edu

**Gary Cheng**[*]
Stanford University
chenggar@stanford.edu

**John C. Duchi**
Stanford University
jduchi@stanford.edu

## Abstract

We extend the Approximate-Proximal Point (APROX) family of model-based methods for solving stochastic convex optimization problems, including stochastic subgradient, proximal point, and bundle methods, to the minibatch setting. To do this, we propose two minibatched algorithms for which we prove a non-asymptotic upper bound on the rate of convergence, revealing a linear speedup in minibatch size. In contrast to standard stochastic gradient methods, these methods may have linear speedup in the minibatch setting even for non-smooth functions. Our algorithms maintain the desirable traits characteristic of the APROX family, such as robustness to initial step size choice. Additionally, we show improved convergence rates for "interpolation" problems, which (for example) gives a new parallelization strategy for alternating projections. We corroborate our theoretical results with extensive empirical testing, which demonstrates the gains provided by accurate modeling and minibatching.

## 1  Introduction

We develop parallel stochastic approximate proximal point methods (APROX) for solving the stochastic optimization problem

$$\text{minimize } f(x) = \mathbb{E}_P[F(x; S)] = \int_{\mathcal{S}} F(x; s) dP(s) \tag{1}$$

$$\text{subject to } x \in \mathcal{X}.$$

Here the set $\mathcal{S}$ is a sample space, and for each $s \in \mathcal{S}$, the function $F(\cdot; s) : \mathbb{R}^n \to \mathbb{R}$ is a closed convex function, subdifferentiable on the closed convex set $\mathcal{X} \subset \mathbb{R}^n$. While stochastic gradient methods (SGM) are the de facto choice for problem (1) [32, 22, 8, 29]—enjoying several convergence guarantees [32], with straightforward parallel extensions that make them practically attractive [18, 11, 13]—they are very sensitive to the objective $f$, noise, and hyperparameter tuning [19, 1, 2]. For example, stochastic gradient methods are extremely sensitive to stepsize, and they may even diverge for objectives that do not satisfy their convergence criteria [2].

Motivated by these limitations, several researchers [6, 16, 10, 12, 2] have developed stochastic (approximate) proximal-point and model-based methods as a more robust alternative to standard gradient methods. These APROX methods, as we explain more carefully in Section 1.1, construct a model of the function at each iterate and update by minimizing a regularized version of the model.

---

[*]Denotes equal contribution; authors listed in alphabetical order

Recent results demonstrate the improvements of these frameworks over standard stochastic gradient methods, as the methods demonstrate robustness to stepsize choice, are adaptive to problem difficulty, and converge on a broader range of problems than the stochastic gradient method [12, 2]. Yet these APROX methods are inherently sequential, and as we hit physical limits on processor speeds, it is becoming clear that opportunities for improvements in large-scale computation and energy use must focus on parallelization [14]; it is not immediately apparent how to efficiently parallelize stochastic model-based and proximal point methods.

**Contributions** Motivated by these advantages of APROX over SGM and the importance of parallel computation in large scale stochastic optimization, we propose extensions of APROX to the minibatch setting that allow fast parallelization. We show that our extensions maintain the desirable robustness properties of APROX and enjoy significant speedups in the minibatch size. In contrast to standard optimization methods, which can only guarantee speedups for smooth functions [18, 11], we show that our APROX extensions have speedups even for non-smooth and non-Lipschitz functions assuming a weak form of strong convexity. Moreover, these algorithms exhibit speedup robust to stepsize choices, unlike naive stochastic gradient methods, which require careful stepsize tuning to achieve the desired speedup. Finally, we show on a class of *interpolation problems* [4, 5] that minibatched APROX algorithms enjoy a linear convergence that also improves linearly with minibatch size. Our experimental investigation illustrates the importance of our APROX extensions over standard stochastic gradient methods. Please visit `github.com/garyxcheng/parallel-aprox` for the code for our methods and experiments.

## 1.1 Preliminaries

The starting point of our methods is the APROX framework [10, 12, 2]. The APROX algorithms rely on creating *models* of the function $F$, where the model $F_x$ of $F$ at $x$ satisfies the conditions

(C.i) The function $y \mapsto F_x(y; s)$ is convex and subdifferentiable on $\mathcal{X}$.

(C.ii) The model $F_x$ satisfies the equality $F_x(x; s) = F(x; s)$ and $F_x(y; s) \leq F(y; s)$ for all $y$.

With said model, at iterate $k$, APROX algorithms perform the update

$$x_{k+1} \coloneqq \operatorname*{argmin}_{x \in \mathcal{X}} \left\{ F_{x_k}(x; S_k) + \frac{1}{2\alpha_k} \|x - x_k\|_2^2 \right\}. \tag{2}$$

The APROX framework builds on the idea that using better models in the update (2) results in more stable algorithms with better guarantees. The following models are key to illustrating our methods:

- *Stochastic gradient methods:* for some $F'(x; s) \in \partial F(x; s)$, use the linear model

$$F_x(y; s) \coloneqq F(x; s) + \langle F'(x; s), y - x \rangle. \tag{3}$$

- *Proximal point methods:* use the full proximal model

$$F_x(y; s) \coloneqq F(y; s). \tag{4}$$

- *Truncated methods:* for some $F'(x; s) \in \partial F(x; s)$, use a truncated version of the function

$$F_x(y; s) \coloneqq \max \left\{ F(x; s) + \langle F'(x; s), y - x \rangle, \inf_{z \in \mathcal{X}} F(z; s) \right\}. \tag{5}$$

The truncated model (5) is often easy to apply. In most machine learning applications, loss functions are non-negative, so $F$ is readily modeled by $F_{x_k}(x; s) = [F(x_k; s) + \langle F'(x_k; s), x - x_k \rangle]_+$. We note that it is possible to use a lower bound instead of the infimum in (5). The full proximal (4) and truncated (5) models provide more accurate approximations of $F$ than the linear model (3); which Asi and Duchi [2, 1], motivated by previous work in the area [16, 15, 10], show yields more robust algorithms with better theoretical and practical convergence.

**Notation** For a convex function $f$, $\partial f(x)$ denotes its subgradient set at $x$, and $f'(x) \in \partial f(x)$ denotes an arbitrary element of the subdifferential. We let $\mathcal{X}^\star = \operatorname{argmin}_{x \in \mathcal{X}} f(x)$ denote the set of minimizers for problem (1) and $x^\star \in \mathcal{X}^\star$ denote a single minimizer. We let $\mathcal{F}_k \coloneqq \sigma(S_1, \ldots, S_k)$ be the $\sigma$-field generated by the first $k$ random variables $S_i$, so $x_k \in \mathcal{F}_{k-1}$ for all $k$ under iteration (2). We defer proofs to the appendix.

## 1.2 Related work

Stochastic gradient methods [26] are the most widely used method for solving stochastic minimization problems; an enormous literature exists [24, 25, 32, 22, 31, 17, 3]. Several researchers also demonstrate the improvements that minibatching and parallelization can provide stochastic gradient methods, which can enjoy linear speedups as batch sizes increase [18, 11, 13, 23, 9]. Other works recognize the instability of stochastic gradients methods and importance of robustness, demonstrating situations where they can have slow convergence as a result of mis-specified stepsizes [22, 3, 1].

Rockafellar [27] introduces proximal point methods, which have seen a resurgence in applications to stochastic optimization [16, 6, 15, 7, 20]. Of most relevance to our work are extensions of the stochastic proximal methods that use approximate models in the proximal update [12, 10, 2, 1]. Asi and Duchi [2] develop a stochastic approximate proximal point method, namely APROX, and establish several convergence guarantees and stability properties that are superior to standard stochastic gradient methods. Yet this work does not address our central challenge: how to leverage parallelization.

## 2 Methods

While it is usually clear how to extend standard stochastic gradient methods to parallel settings, the same is not true for proximal methods as there are many different ways to model the function or average the iterates. Indeed, in this section, we describe three different methods for extending APROX—all of which are identical for linear models (SGM)—but can exhibit different behavior as reflected by our experiments. Given a choice of model $F_x(\cdot; s)$ and $m$ samples $S_k^{1:m} \in \mathcal{S}^m$, we propose three methods.

**Iterate averaging** (*IA*): The most natural and direct way to extend APROX to the minibatch setting is to average the individual updates per sample:

$$x_{k+1} = \frac{1}{m} \sum_{i=1}^{m} x_{k+1}^i \quad \text{where} \quad x_{k+1}^i := \operatorname*{argmin}_{x \in \mathcal{X}} \left\{ F_{x_k}(x; S_k^i) + \frac{1}{2\alpha_k} \|x - x_k\|_2^2 \right\}. \quad (6)$$

This method's simplicity and (near) full parallelization makes it attractive. Its empirical performance, however, is weaker than our other proposed methods' performance.

Instead of averaging the iterates, we propose an alternative algorithm: at each iteration, perform the update

$$x_{k+1} := \operatorname*{argmin}_{x \in \mathcal{X}} \left\{ \overline{F}_{x_k}(x; S_k^{1:m}) + \frac{1}{2\alpha_k} \|x - x_k\|_2^2 \right\}, \quad (7)$$

where $\overline{F}_{x_k}(x; S_k^{1:m})$ is a model of the average function

$$\overline{F}(x; S_k^{1:m}) = \frac{1}{m} \sum_{i=1}^{m} F(x; S_k^i) \quad (8)$$

satisfying Conditions (C.i) and (C.ii). While our theorems hold generally for any $\overline{F}_{x_k}(x; S_k^{1:m})$ satisfying Conditions (C.i) and (C.ii), we now present two instances of the model family which are of practical interest.

**Truncated Average** (*TruncAv*): The simplest such model—beyond the naive linear model—is an extension of the truncated model (5). Here, we set

$$\overline{F}_x(y; S^{1:m}) := \max \left\{ \overline{F}(x; S^{1:m}) + \langle \overline{F}'(x; S^{1:m}), y - x \rangle, \inf_{z \in \mathcal{X}} \overline{F}(z; S^{1:m}) \right\}.$$

In the standard case that the functions $F$ are nonnegative, for $\overline{g}_k := \overline{F}'(x; S_k^{1:m})$, the update (7) is

$$x_{k+1} = x_k - \min \left\{ \alpha_k, \frac{\overline{F}(x_k; S_k^{1:m})}{\|\overline{g}_k\|_2^2} \right\} \overline{g}_k. \quad (9)$$

The update (9) for the truncated models thus yields an embarrassingly parallelizable scheme: each worker computes $F(x_k; S_k^i)$ and $F'(x_k; S_k^i)$, which need only be averaged to apply the update (9).

**Average of Truncated Models** (*AvTrunc*): The update (9) ignores some structural aspects of the objectives $F$; it is natural to consider an even more accurate model, which averages models of individual samples. Here we construct a model $F_{x_k}(x; S_k^i) = \max\{F(x; S_k^i) + \langle F'(x; S_k^i), y - x \rangle, \inf_{z \in \mathcal{X}} F(z; S_k^i)\}$ for each sample, noting that $\frac{1}{m} \sum_{i=1}^m F_{x_k}(\cdot; S_k^i)$ satisfies conditions (C.i) and (C.ii), then set

$$x_{k+1} := \operatorname*{argmin}_{x \in \mathcal{X}} \left\{ \frac{1}{m} \sum_{i=1}^m F_{x_k}(x; S_k^i) + \frac{1}{2\alpha_k} \|x - x_k\|_2^2 \right\}. \tag{10}$$

There is an $O(m)$-dimensional dual problem to (10), which allows this method to be practically parallelizable. Consider the equivalent iterative algorithm

$$\lambda_k = \operatorname*{argmax}_{\lambda} \max_{\nu} -\frac{\alpha_k}{2} \| \sum_{i=1}^m \lambda_{(i)} F'(x; S_k^i) \|_2^2 + \sum_{i=1}^m \lambda_{(i)} F(x_k; S_k^i)$$

$$\text{subject to } \frac{1}{m} + \lambda_{(i)} - \nu_{(i)} = 0, \nu \geq 0$$

$$x_{k+1} = x_k - \sum_{i=1}^m \lambda_{k,(i)} F'(x; S_k^i),$$

where $\lambda_{(i)}$ is the $i$th component of the vector $\lambda$.

**Remark**     We note that we could easily generalize TruncAv and AvTrunc methods beyond the truncated model. Let $\mathcal{H}$ denote the set of convex functions. Let $G : \mathcal{H} \times \mathcal{X} \to \mathcal{H}$ be some operator which takes in a convex function $F$ and a point $y$ in the domain as input and constructs a model of $F$ at $y$ satisfying Conditions (C.i) and (C.ii). Then the TruncAv model is more generally written as $G(\overline{F}(x; S^{1:m}), x_k)$. The AvTrunc model is more generally $\sum_{i=1}^m G(F(x; S^i), x_k)/m$.

Before proceeding to our theoretical guarantees, we provide a simple example to illustrate the differences between these methods. Consider the problem of finding a point in the intersection of convex sets $C_1, \ldots, C_k$ by minimizing $f(x) = k^{-1} \sum_{i=1}^k \operatorname{dist}(x, C_i)$. Let $m = 2$, and consider the truncated model (5), which exhibits different behavior for each parallelization strategy. Figure 1 illustrates the IA, TruncAv, and AvTrunc updates given infinite stepsize (which still guarantees convergence if there is $x^\star \in \cap_i C_i$ [2]). In this case, iterate averaging (6) projects the current iterate $x_k$ to the two sets in the batch and averages these updates. TruncAv (9) constructs an (average) hyperplane that gives a (somewhat) good representation of the two sets and projects to this set. AvTrunc (10) provides a more accurate representation of the two sets using two hyperplanes and projects to this set.

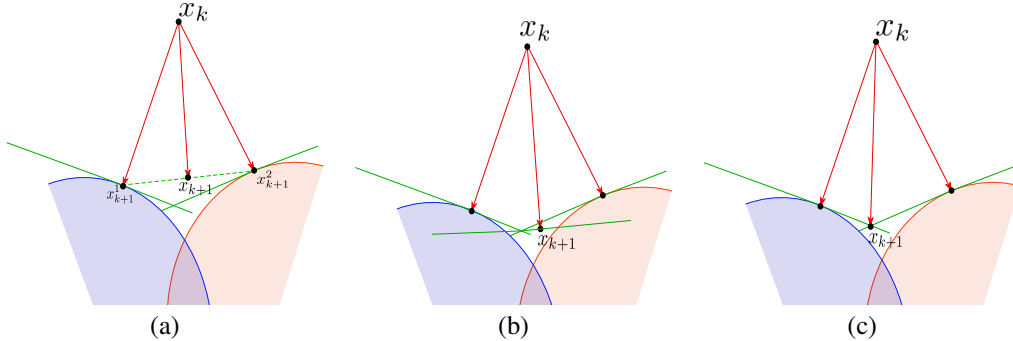

| (a) | (b) | (c) |

**Figure 1:**   Updates for (a) IA (6) with truncated model, (b) TruncAv (9), and (c) AvTrunc (10).

## 3   Non-asymptotic convergence guarantees and minibatch speedups

Having described our methods, we prove several convergence guarantees and minibatch speedups for these methods. We begin in Section 3.1 with standard speedup result—which resembles known results for SGM—which holds for any model satisfying Conditions (C.i) and (C.ii). In Section 3.2 we show that, somewhat surprisingly, our methods using the full proximal model (4) enjoy linear speedup in minibatch size even for non-smooth functions.

## 3.1 Speedup for smooth functions

We begin by proving minibatch speedup results for smooth functions. As is the case for stochastic gradient methods [11], our results require a bound on the variance of $\nabla F(x_k; S_k)$. We therefore begin with a one-step progress guarantee for any APROX model that depends on the noise of the estimate of $\nabla f(x)$.

**Lemma 3.1.** *Let $f(\cdot)$ be convex and have L-Lipschitz gradient. Define the function value errors $e_k = [F_{x_k}(x^\star; S_k) - f(x^\star)] - [F(x_k, S_k) - f(x_k)]$ and let $\alpha_k = \frac{1}{L+\eta_k}$. Let $x_{k+1}$ be generated using the update (2) using any model $F_{x_k}(x; S_k)$ satisfying (C.i) and (C.ii). Then*

$$f(x_{k+1}) - f(x^\star) \le \frac{1}{2\alpha_k}\left[\|x_k - x^\star\|_2^2 - \|x_{k+1} - x^\star\|_2^2\right] + e_k + \frac{1}{2\eta_k}\|\nabla F(x_k; S_k) - \nabla f(x_k)\|_2^2.$$

By noting that $\mathbb{E}[e_k] \le 0$, Lemma 3.1 nearly immediately implies the following theorem.

**Theorem 1.** *Let the conditions of Lemma 3.1 hold and $\eta_k$ be non-decreasing. Additionally, assume that $\mathrm{diam}(\mathcal{X}) = \sup_{x,x'\in\mathcal{X}} \|x - x'\|_2 \le R$ and $\mathbb{E}[\|\nabla F(x; S) - \nabla f(x)\|_2^2] \le \sigma_0^2$ for $x \in \mathcal{X}$. Then*

$$\sum_{i=1}^{k} \mathbb{E}[f(x_{i+1}) - f(x^\star)] \le \frac{LR^2}{2} + \frac{R^2\eta_k}{2} + \sum_{i=1}^{k}\frac{\sigma_0^2}{2\eta_i}.$$

Having established a convergence result that depends on the noise of the gradient estimates, a speedup guarantee for the minibatched APROX methods is nearly immediate. Equation (7) applies APROX updates to the average (8), reducing variance by a factor of $m$, giving

**Corollary 3.1.** *Let the conditions of Theorem 1 hold, let $\eta_k = \eta_0\sqrt{k}$ with $\eta_0 = \frac{\sigma_0}{\sqrt{m}R}$, and let $x_k$ be generated using (7) with any model satisfying (C.i) and (C.ii) with minibatch $m$. Then the average $\overline{x}_k = \frac{1}{k}\sum_{i=2}^{k+1} x_i$ satisfies*

$$\mathbb{E}[f(\overline{x}_k) - f(x^\star)] \le \frac{LR^2}{2k} + \frac{3R\sigma_0}{2\sqrt{km}}.$$

When the batch size $m \ll \frac{k\sigma_0^2}{L^2R^2}$, the second term dominates the rate of convergence. Letting $T(\epsilon)$ denote the number of iterations to achieve $\mathbb{E}[f(\overline{x}_{T(\epsilon)}) - f(x^\star)] \le \epsilon$, we obtain that $T(\epsilon) \lesssim \frac{R^2\sigma_0^2}{\epsilon^2 m}$, that is, there is a linear speedup as a function of the minibatch of size $m$. This is similar to the speedup that standard stochastic gradient methods achieve [18, 11], and it is minimax optimal.

## 3.2 Speedup for non-smooth functions

We turn now to a complementary look at potentially non-Lipschitz and non-smooth functions, studying the minibatch APROX framework (7) using the full proximal model (4), that is, a minibatched stochastic proximal point method. Here, we show a speedup as the minibatch size grows, but in contrast to the previous section, it has the benefit that it requires only that the noise of the gradient estimate is bounded over the optimal set $\mathcal{X}^\star$. Here, we consider the following restricted strong convexity (RSC) assumption [2].

**Assumption A1** (Restricted strong convexity)**.** *The functions $F(\cdot; s)$ are strongly convex w.r.t. the matrix $\Sigma(s) \succeq 0$, i.e., for arbitrary x and y, $F(y; s) \ge F(x; s) + F'(x, s)^T(y - x) + (1/2)(y - x)^T\Sigma(s)(y - x)$, for all $F'(x, s) \in \partial F(x; s)$. The matrix $\Sigma(S)$ satisfies $\mathbb{E}[\Sigma(S)] \succeq \lambda_{\min}I_{n\times n}$ with $\lambda_{\min} > 0$.*

Fixing an (otherwise arbitrary) constant $c > 1$, we define

$$\overline{\Sigma}_k = \mathbb{E}\left[\frac{\Sigma(S)}{1 + 2\alpha_k\lambda_{\max}(\Sigma(S))}\right] \text{ and } \lambda_k = \lambda_{\min}(\overline{\Sigma}_k).$$

We then have the following convergence guarantee for minibatched stochastic proximal point methods (4), which is a consequence of Proposition 5 of Asi and Duchi [2].

**Corollary 3.2.** *Let Assumption A1 hold and assume $\mathbb{E}[\|F'(x^\star; S)\|_2^2] \leq \sigma_1^2$. For $\alpha_k = \alpha_0 k^{-\beta}$ with $\beta \in (0, 1)$, the APROX method with proximal point model (4) using batch size $m$ guarantees*

$$\mathbb{E}[\|x_{k+1} - x^\star\|_2^2] \leq \exp\left(-\lambda_0 \sum_{i=1}^{k} \alpha_i\right) \|x_1 - x^\star\|_2^2 + C \cdot \frac{\sigma_1^2}{m\lambda_0} \alpha_k \cdot \log k,$$

*where $C$ is a numerical constant and $\lambda_0 = \lambda_{\min}(\overline{\Sigma}_0)$.*

The second term dominates the convergence rate in Corollary 3.2, so $\alpha_k = b/k$ for a large constant $b$, we obtain $\mathbb{E}[\|x_{k+1} - x^\star\|_2^2] \leq O(\frac{\sigma_1^2 \log k}{mk\lambda_0})$ demonstrating the desired speedup in $m$. Existing results for standard stochastic gradient methods do not exhibit a speedup in the batch size for non-smooth functions [11].

## 4 Linear rates and speedups for interpolation problems

The previous section shows that minibatching produces speedups for APROX for certain families of functions. Here, we consider a specific class of interpolation problems, where we will show linear convergence rates and minibatch speedups for APROX.

**Definition 4.1.** *The optimization problem (1) is an interpolation problem if there exists $x^\star \in \mathcal{X}^\star := \operatorname{argmin}_{x \in \mathcal{X}} f(x)$ such that for $P$-almost all $s \in \mathcal{S}$, we have $\inf_{x \in \mathcal{X}} F(x; s) = F(x^\star; s)$.*

While this restricts the class of objectives, many problems satisfy this condition [21, 28, 2], including overdetermined linear systems, finding a point in the intersection of convex sets, and the modern machine learning problems where it is possible to achieve zero training loss [4, 5]. For interpolation problems, we consider bounding the noise of the gradient estimate by its suboptimality.

**Assumption A2.** *There is $\sigma_2^2 < \infty$ such that for every $x \in \mathcal{X}$, $\mathbb{E}[\|\nabla f(x) - \nabla F(x; S)\|_2^2] \leq \sigma_2^2 \operatorname{dist}(x, \mathcal{X}^\star)^2$.*

Assumption A2 holds for many interpolation problems. For example, consider a linear regression problem with data $s = (a, b) \in \mathbb{R}^n \times \mathbb{R}$, $a^T x^\star = b$ for all $(a, b)$, and $F(x; (a, b)) = \frac{1}{2}(a^T x - b)^2$. We immediately have $\operatorname{Var}(\nabla F(x; S)) \leq \mathbb{E}[\|a\|_2^2 \langle a, x - x^\star \rangle^2]$, evidently satisfying Assumption A2. The assumption implies linear convergence guarantees:

**Proposition 1.** *Assume $f$ is $\lambda$-strongly convex, has $L$-Lipshitz gradients, and satisfies Assumption A2 and Def. 4.1. Let $x_k$ be generated using update (7) using any model satisfying (C.i) and (C.ii) with minibatch $m > \sigma_2^2/(\lambda(L + \lambda))$. Set $\alpha_k = (L + \eta)^{-1}$. For $\eta = \max\left\{L, \frac{8\sigma_2^2}{m\lambda}\right\}$, we have*

$$\mathbb{E}[\operatorname{dist}(x_k, \mathcal{X}^\star)^2] \leq \exp\left(-k \min\left\{\frac{\lambda}{8L}, \frac{m\lambda^2}{64\sigma_2^2}\right\}\right) \mathbb{E}[\operatorname{dist}(x_0, \mathcal{X}^\star)^2].$$

It is possible to relax the strong convexity condition on $f$ into a weaker quadratic growth assumption:

**Assumption A3** (Quadratic Growth). *There exist $\lambda_0, \lambda_1 > 0$ such that for all $x \in \mathcal{X}$ and $\alpha > 0$,*

$$(f(x) - f(x^\star)) \min\left\{\alpha, \frac{f(x) - f(x^\star)}{\|f'(x)\|^2}\right\} \geq \min\{\lambda_0 \alpha, \lambda_1\} \operatorname{dist}(x, \mathcal{X}^\star)^2.$$

Functions satisfying Assumption A3 need only grow quadratically away from their minimizers, so they are less restrictive than a global strong convexity assumption. We then have the following.

**Theorem 2.** *Assume $f$ has $L$-Lipschitz gradients and satisfies Assumptions A2, A3, and Definition 4.1. Define the step sizes $\alpha_k = (2L + 2\eta_k)^{-1}$ where $\sum_{i=1}^{\infty} \eta_i^{-1} = \infty$ and $\sum_{i=1}^{\infty} \eta_i^{-2} < \infty$, and let $K$ be the largest value such that for all $k > K$, $\alpha_k \leq \lambda_1/2\lambda_0$. If $x_k$ is generated using update (7) using any model satisfying (C.i) and (C.ii) with minibatch $m$, then*

$$\mathbb{E}[\operatorname{dist}(x_k, \mathcal{X}^\star)^2] \leq \exp\left(-(K \wedge k)\lambda_1/2 - \sum_{i=K\wedge k+1}^{k} \lambda_0 \alpha_k + \sum_{i=1}^{k} \frac{\sigma_2^2}{m(L + \eta_i)\eta_i}\right) \operatorname{dist}(x_0, \mathcal{X}^\star)^2.$$

To give some intuition for this bound, we provide a heuristic sketch for stepsizes with a given minibatch size $m$, assuming for simplicity that $\lambda_0 = \lambda_1 = 1$. Take $\eta_k = \eta_0 k^\beta / \sqrt{m}$, where $\beta \in (1/2, 1)$. Then $\sum_{i=1}^k \frac{\sigma_2^2}{m(L+\eta_i)\eta_i} \lesssim \frac{\sigma_2^2}{\eta_0^2}$, and $K = (\sqrt{m}/\eta_0)^{1/\beta}$, while for large $k$ we have $\sum_{i=K+1}^k \alpha_i \gtrsim k^{1-\beta}\sqrt{m}/\eta_0^{1-\beta}$. This gives a heuristic convergence guarantee of $\mathbb{E}[\mathrm{dist}(x_k, \mathcal{X}^\star)^2] \lesssim \exp(-k^{1-\beta}\sqrt{m}/\eta_0^{1-\beta} + \sigma_2^2/\eta_0^2)\,\mathrm{dist}(x_0, \mathcal{X}^\star)^2$, which (taking the limit as $\beta \downarrow \frac{1}{2}$) gives $\mathbb{E}[\mathrm{dist}(x_k, \mathcal{X}^\star)^2] \lesssim \exp(-\sqrt{km}/\eta_0)\,\mathrm{dist}(x_k, \mathcal{X}^\star)^2$, exhibiting linear speedup in $m$.

## 5   Experiments

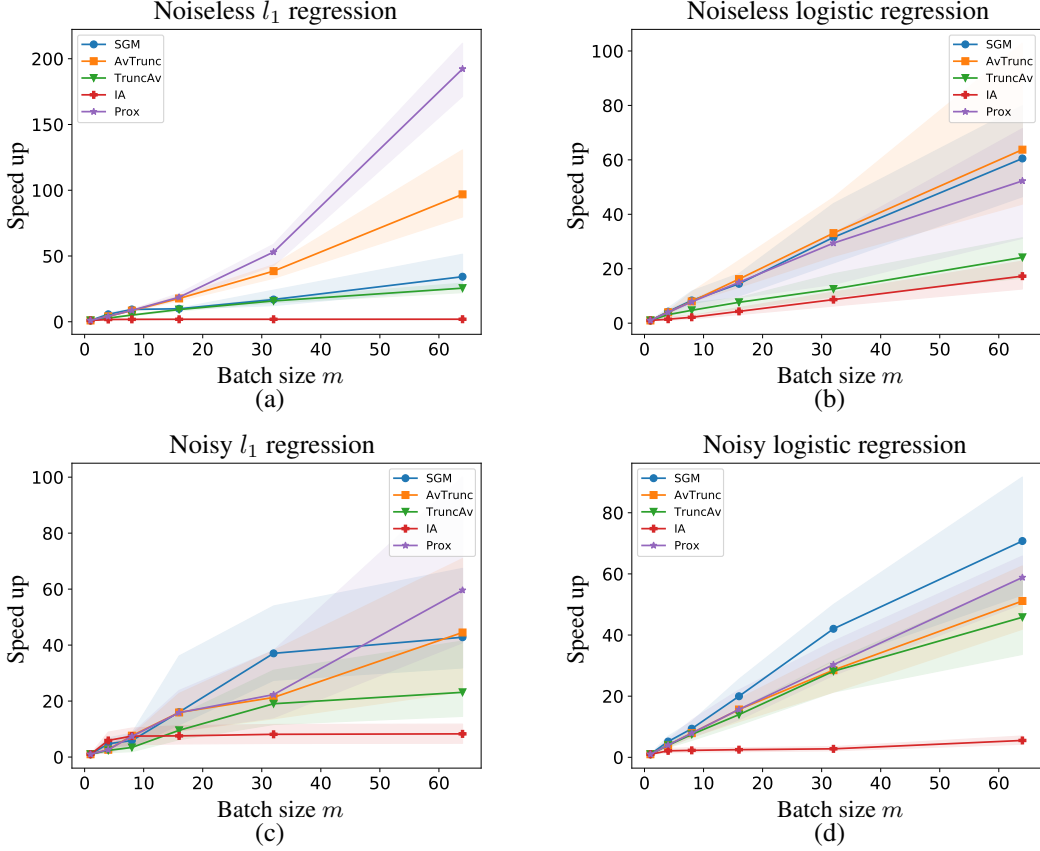

**Figure 2:**   Speed ups with best possible stepsizes vs. batch size

Our goal now is to demonstrate the speedup and robustness of APROX methods with minibatches, comparing the relative performance of the proposed methods. We compare

1. SGM: stochastic gradient methods, i.e., the linear model (3).

2. Proximal: uses the full proximal model (4) with averaged function (8) (Prox).

3. Truncated-IA: the truncated model (5) in iterate averaging (6) (IA).

4. TruncAv (9).

5. AvTrunc (10).

We study performance on linear, absolute loss, and logistic regression. In each case, we have data in the form of a matrix $A \in \mathbb{R}^{n \times d}$. We have $b \in \mathbb{R}^n$ for linear and absolute loss regression, and $b \in \{-1, +1\}^n$ for logistic regression. We use $n = 1000$, $d = 40$ minibatch sizes $m \in \{1, 4, 8, 16, 32, 64\}$ and initial stepsizes $\alpha_0 \in \{10^{-2}, 10^{-1.5}, \dots, 10^{2.5}, 10^3\}$ ($\alpha_0 \in \{10^{-2}, 10^{-1.5}, \dots, 10^{4.5}, 10^5\}$ for logistic regression). For all experiments we run 30 trials

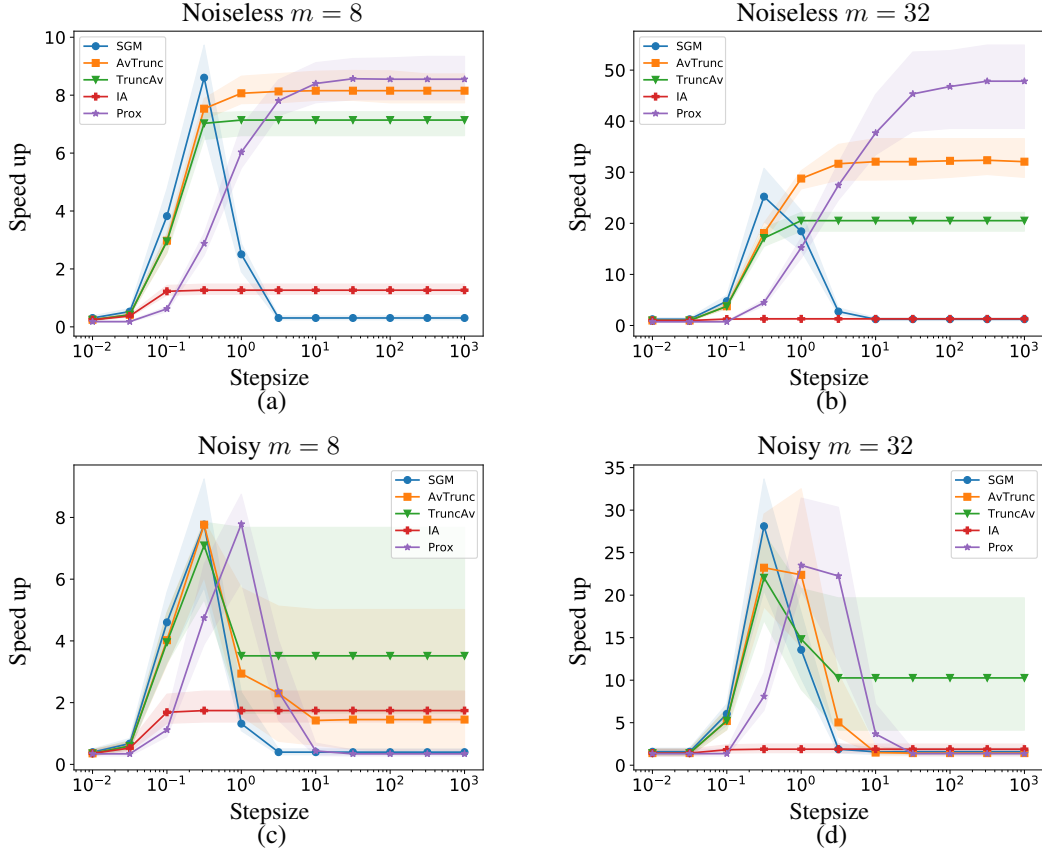

**Figure 3:** Speed up vs. stepsizes for linear regression

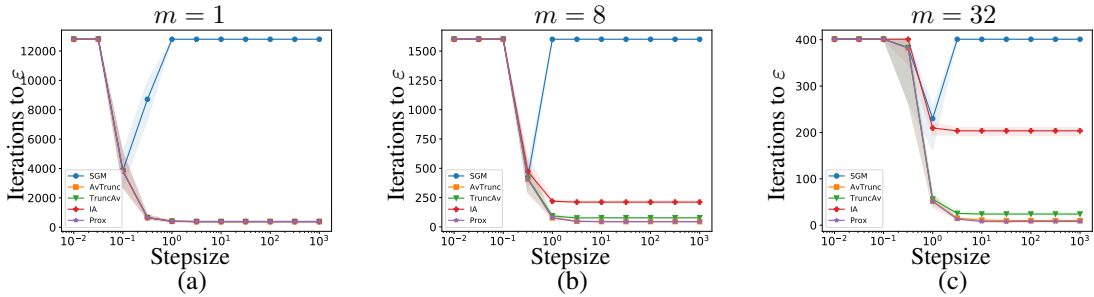

**Figure 4:** Time to convergence vs. stepsizes for noiseless absolute regression

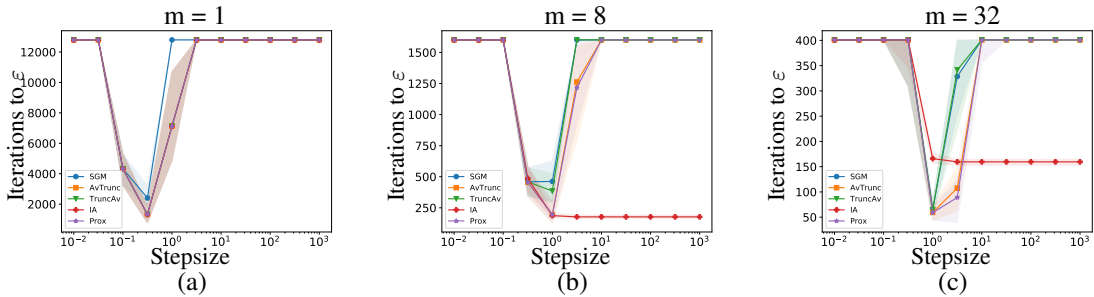

**Figure 5:** Time to convergence vs. stepsizes for absolute regression (noisy with $\sigma = 0.5$)

with different seeds and plot the $95\%$ confidence sets. We describe the objective function and noise adding mechanism for each experiment below.

    a) Linear Regression: We have $f(x) = \frac{1}{2n}\|Ax - b\|_2^2$. For each experiment we first generate a random matrix $A$ and $x^\star \sim \mathsf{N}(0, I_d)$ and then use $b = Ax^\star + \sigma v$ with $v \sim \mathsf{N}(0, I_n)$. In the noisy setting for our experiments, we set $\sigma = 0.5$, we get different noise levels.

    b) Absolute loss regression: We have $f(x) = \frac{1}{2n}\|Ax - b\|_1$. We generate a random matrix $A$, $x^\star \sim \mathsf{N}(0, I_d)$ and use $b = Ax^\star + v$, where $v_i \sim \mathrm{Lap}(0, \frac{\sigma}{2})$ ($\mathrm{Lap}(\cdot, \cdot)$ denotes the Laplace distribution). In the noisy setting for our experiments, we set $\sigma = 0.5$, we get different noise levels.

    c) Logistic Regression: We have $f(x) = \frac{1}{2n}\sum_{i=1}^n \log(1 + \exp(-b_i\langle a_i, x\rangle))$. We generate a random matrix $A$, $x^\star \in \mathsf{N}(0, I_d)$ and $b = \mathrm{sign}(\langle a_i, x\rangle)$. To add noise, we flip each sign in $b$ independently with probability $p$. In the noisy setting for our experiments, we set $p = 0.01$, we get different noise levels.

For the experiments in the main paper, we set the condition number of $A$ to 1. Please refer to the appendix to see experiments where the condition number of $A$ is set to 10.

For each of the problem types, we use stepsizes $\alpha_k = \alpha_0 k^{-1/2}$ and find the number $T_m(\alpha_0)$ of iterations $k$ required to reach $\varepsilon$ accuracy, $f(x_k) - f(x^\star) \le \varepsilon$. A max iteration termination threshold is chosen such that roughly three out of the five methods converge to $\varepsilon$ accuracy before the threshold; if a method does not converge in that time, we set $T_m(\alpha_0)$ to the threshold value. We also let $T_m^\star = \min_{\alpha_0} T_m(\alpha_0)$ denote the smallest time to convergence for a given method using batch size $m$. The supplement contains our full experimental setup. We present three types of plots.

**Best speedups for minibatching** (Fig. 2) For each method, we plot $T_1^\star/T_m^\star$ against the minibatch size $m$ to show the speedup minibatching provides using the best step sizes. This shows the best possible speedup obtained by minibatching through tuning the initial step size $\alpha_0$. All algorithms except the iterate averaging method IA (6) enjoy a linear speedup w.r.t. minibatch sizes for small minibatches before it tapers off in both absolute and logistic regression. In the absolute regression case, the full proximal model performs much better than any other model, while in logistic regression, the behaviour of each proposed model is comparable; this is consistent with our arbitrary acceleration results for proximal-point method (Cor. 3.2), while our results for TruncAv and AvTrunc (Thms. 1 and 2) hold for smooth functions.

**Speedups w.r.t. step-size** (Fig. 3) For each method and minibatch size, we plot $T_1^\star/T_m(\alpha_0)$ against the initial step size $\alpha_0$. This illustrates the robustness of speedups obtained by minibatching to the choice of step size $\alpha_0$. In Fig. 3, we expect the stochastic gradient method to exhibit improved performance for *good* stepsizes, though this improvement is not robust: it achieves no speedup when stepsizes are poorly chosen. For the parallel APROX methods (AvTrunc and TruncAv), however, we observe that the speedup persists over many stepsizes, essentially perfectly in the noiseless case and with some degradation in the noisy case.

**Time to solution w.r.t. step-size:** (Fig. 4 and Fig. 5) For each method and minibatch size, we plot $T_m(\alpha_0)$ against the initial step size $\alpha_0$. Whereas the previous plots explore relative performance, this plot characterizes performance on an absolute scale. In the noiseless case (Fig. 4), we clearly observe the robustness of model-based methods compared to SGM. IA performs better than SGM, but worse than others; TruncAv, AvTrunc, and the full prox method have similar performance. In the noisy case, surprisingly, IA is more robust to stepsizes, though the performance under the best stepsize is not as good as AvTrunc or TruncAv. We believe this is a consequence of extremely conservative updates. A natural next direction for future work here is to understand precisely which updates can yield improvement with parallelization.

## Broader Impact

Data centers draw increasing amounts of the total energy we consume, and increasing applications of machine learning mean that model-fitting and parameter exploration require a larger and larger proportion of their energy expenditures [1, 14, 30]. Indeed, as Asi and Duchi [1] note, the energy to train and tune some models is roughly on the scale of driving thousands of cars from San Francisco to Los Angeles, while training a modern transformer network (with architecture search) generates roughly six times the total $CO_2$ of an average car's lifetime [30]. It is thus centrally important to build more efficient and robust methods, which allow us to avoid wasteful hyperparameter search but simply work.

A major challenge in building better algorithms is that fundamental physical limits have forced CPU speed and energy to essentially plateau; only by parallelization can we harness both increasing speed and reduce the energy to fit models [14]. In this context, our methods take a step toward reducing the energy and overhead to perform machine learning.

Taking a step farther back, we believe optimization and model-fitting research in machine learning should refocus its attention: rather than developing algorithms that, with appropriate hyperparameter tuning, achieve state-of-the-art accuracy for a given dataset, we should evaluate algorithms by whether they *robustly* work. This would allow a more careful consideration of an algorithms' costs and benefits: is it worth $2\times$ faster training, for appropriate hyperparameters, if one has to spend $25\times$ as much time to *find* the appropriate algorithmic hyperparameters? Even more, as Strubell et al. [30] point out, the extraordinary costs of hyperparameter tuning for fitting large-scale models price many researchers out of making progress on certain frontiers; to the extent that we can mitigate these challenges, we will allow more equity in who can help machine learning progress.

## Funding Transparency Statement

Funding in direct support of this work: NSF CAREER CCF-1553086, ONR YIP N00014-19-2288, Sloan Foundation, NSF HDR 1934578 (Stanford Data Science Collaboratory), Stanford DAWN Consortium, and the Professor Michael J. Flynn Stanford Graduate Fellowship.

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
