[Supplementary Material]

# Appendix

In Appendix A, we provide proofs for all the theorems, lemmas, and propositions stated in the main body. In Appendix B, we provide additional experiments supporting our claims.

## A  Proofs

**Lemma A.1.** *(Fenchel Young inequality) For any convex function $f(\cdot)$, let $f^*(\cdot)$ denote its Fenchel dual. Then, we have*

$$f(x) + f^*(y) \geq x^T y. \tag{11}$$

Specifically, we use the following two instances of Fenchel-Young extensively:

1. For $f(x) = \frac{\|x\|_2^2}{2\beta}$, we have $f^*(y) = \frac{\beta}{2} \|y\|_2^2$ for any constant $\beta > 0$. Thus, we obtain

$$x^T y \leq \frac{1}{2\beta} \|x\|_2^2 + \frac{\beta}{2} \|y\|_2^2.$$

2. For $f(x) = \frac{x^T M x}{2}$, we have $f^*(y) = \frac{y^T M^{-1} y}{2}$ for any positive definite matrix $M$. Thus, we have

$$x^T y \leq \frac{x^T M x}{2} + \frac{y^T M^{-1} y}{2}.$$

### A.1  Proof of Lemma 3.1

We begin with Lemma 3.3 of [2], which gives

$$\frac{1}{2} \|x_{k+1} - x^\star\|_2^2 \leq \frac{1}{2} \|x_k - x^\star\|_2^2 - \alpha_k [F_{x_k}(x_{k+1}; S_k) - F_{x_k}(x^\star; S_k)] - \frac{1}{2} \|x_{k+1} - x_k\|_2^2. \tag{12}$$

Now, let $g_k := \nabla F(x_k; S_k)$ and define $\xi_k := g_k - \nabla f(x_k)$. Using convexity of $F_{x_k}(\cdot; S_k)$, we have $F_{x_k}(x_{k+1}; S_k) \geq F_{x_k}(x_k; S_k) + \langle g_k, x_{k+1} - x_k \rangle$ and thus

$$F(x^\star; S_k) - F_{x_k}(x_{k+1}; S_k) \leq F_{x_k}(x^\star; S_k) - F(x_k; S_k) + \langle \nabla f(x_k), x_k - x_{k+1} \rangle$$
$$+ \langle \xi_k, x_k - x_{k+1} \rangle.$$

Applying the definition of error $e_k := [F_{x_k}(x^\star; S_k) - f(x^\star)] - [F(x_k; S_k) - f(x_k)]$ and using the fact that $f$ is $L$-smooth, we get

$$F(x^\star; S_k) - F_{x_k}(x_{k+1}; S_k) \leq f(x^\star) - f(x_{k+1}) + \frac{L}{2} \|x_k - x_{k+1}\|_2^2 + e_k + \langle \xi_k, x_k - x_{k+1} \rangle.$$

Substituting in (12) and rearranging,

$$f(x_{k+1}) - f(x^\star) \leq \frac{1}{2\alpha_k} [\|x_k - x^\star\|_2^2 - \|x_{k+1} - x^\star\|_2^2] - \frac{1}{2\alpha_k} \|x_k - x_{k+1}\|_2^2 \tag{13}$$
$$+ e_k + \langle \xi_k, x_k - x_{k+1} \rangle + \frac{L}{2} \|x_k - x_{k+1}\|_2^2.$$

Using Fenchel-Young's inequality on $\langle \xi_k, x_k - x_{k+1} \rangle$,

$$f(x_{k+1}) - f(x^\star) \leq \frac{1}{2\alpha_k} [\|x_k - x^\star\|_2^2 - \|x_{k+1} - x^\star\|_2^2] - \frac{1}{2\alpha_k} \|x_k - x_{k+1}\|_2^2$$
$$+ e_k + \frac{1}{2\eta_k} \|\xi_k\|_2^2 + \frac{L + \eta_k}{2} \|x_k - x_{k+1}\|_2^2$$
$$= \frac{1}{2\alpha_k} \left[ \|x_k - x^\star\|_2^2 - \|x_{k+1} - x^\star\|_2^2 \right] + e_k + \frac{1}{2\eta_k} \|\xi_k\|_2^2.$$

## A.2 Proof of Theorem 1

Summing Lemma 3.1 we get,

$$\sum_{i=1}^{k}[f(x_{i+1}) - f(x^\star)] \leq \frac{1}{2}\sum_{i=2}^{k}\left(\frac{1}{\alpha_i} - \frac{1}{\alpha_{i-1}}\right)\|x_i - x^\star\|_2^2 - \frac{\|x_{k+1} - x^\star\|_2^2}{2\alpha_{k+1}}$$

$$+ \frac{\|x_1 - x^\star\|_2^2}{2\alpha_1} + \sum_{i=1}^{k}e_i + \sum_{i=1}^{k}\frac{1}{2\eta_i}\|\nabla F(x_i; S_i) - \nabla f(x_i)\|_2^2$$

$$\leq \frac{R^2}{2\alpha_k} + \sum_{i=1}^{k}e_i + \sum_{i=1}^{k}\frac{1}{2\eta_i}\|\nabla F(x_i; S_i) - \nabla f(x_i)\|_2^2.$$

After taking expectations and using the facts $\mathbb{E}[e_k] \leq 0$, $\alpha_k = \frac{1}{L+\eta_k}$, we get the stated result.

## A.3 Proof of Proposition 1

Using shorthands as in Lemma 3.1 and assuming $f(x^\star) = 0$, Lemma 3.1 implies:

$$\frac{1}{2}D_{k+1}^2 \leq \frac{1}{2}D_k^2 - \alpha f(x_{k+1}) + \alpha e_k + \frac{\alpha}{2\eta}\|\xi_k\|_2^2 \leq \frac{1}{2}D_k^2 - \frac{\alpha\lambda}{2}D_{k+1}^2 + \alpha e_k + \frac{\alpha}{2\eta}\|\xi_k\|_2^2,$$

where the second inequality follows from using strong convexity on the objective to get the bound $f(x_{k+1}) \geq \frac{\lambda}{2}D_{k+1}^2$. We rearrange and take expectations on both sides to obtain

$$\mathbb{E}[D_{k+1}^2] \leq \underbrace{\frac{1}{\alpha\lambda + 1}}_{\leq\exp(-\alpha\lambda/2)}\underbrace{\left(1 + \frac{\alpha\sigma_2^2}{\eta m}\right)}_{\leq\exp(\frac{\alpha\sigma_2^2}{\eta m})}\mathbb{E}[D_k^2] \leq \exp\left(\frac{-\lambda}{2(L+\eta)} + \frac{\sigma_2^2}{\eta(L+\eta)m}\right)\mathbb{E}[D_k^2],$$

where $\mathbb{E}[\|\xi_k\|_2^2] \leq \frac{\sigma_2^2}{m}\mathbb{E}[D_k^2]$ by Assumption A2. Using $2\max\{L,\eta\} > L + \eta > \eta$, we have

$$\mathbb{E}[D_{k+1}^2] \leq \exp\left(\frac{-\lambda}{4\max\{L,\eta\}} + \frac{\sigma_2^2}{\eta^2 m}\right)\mathbb{E}[D_k^2].$$

## A.4 Proof of Theorem 2

WLOG let $f(x^\star) = 0$. We define $D_k := \operatorname{dist}(x_k, \mathcal{X}^\star)$ and $\xi_k := \nabla F(x_k; S_k) - \nabla f(x_k)$ and $\lambda_k := \min(2\alpha_k, f(x_k)/\|\nabla f(x_k)\|_2^2)$. We begin with equation eq. (13) and apply Lemma A.1

$$\frac{1}{2}D_{k+1}^2 \leq \frac{1}{2}D_k^2 + \alpha_k[f(x^\star) - f(x_{k+1})] - \frac{1}{2}\|x_k - x_{k+1}\|_2^2 + \alpha_k e_k$$

$$+ \frac{\alpha_k}{\eta_k^2}\|\xi_k\|_2^2 + \frac{\alpha_k(L+\eta_k)}{2}\|x_k - x_{k+1}\|_2^2.$$

Choosing $\alpha_k = \frac{1}{2}(L+\eta_k)^{-1}$, we obtain

$$\frac{1}{2}D_{k+1}^2 \leq \frac{1}{2}D_k^2 - \alpha_k f(x_{k+1}) - \frac{1}{4}\|x_k - x_{k+1}\|_2^2 + \alpha_k e_k + \frac{\alpha_k}{2\eta_k}\|\xi_k\|_2^2.$$

Observe that due to convexity and our assumption that $f(x^\star) = 0$, we have $f(x_{k+1}) \geq [f(x_k) + \langle\nabla f(x_k), x_{k+1} - x_k\rangle]_+$, meaning

$$\frac{1}{2}D_{k+1}^2 \leq \frac{1}{2}D_k^2 - (\alpha_k[f(x_k) + \langle\nabla f(x_k), x_{k+1} - x_k\rangle]_+ + \frac{1}{4}\|x_k - x_{k+1}\|_2^2) + \alpha_k e_k + \frac{\alpha_k}{2\eta_k}\|\xi_k\|_2^2$$

$$\leq \frac{1}{2}D_k^2 - \inf_y(\alpha_k[f(x_k) + \langle\nabla f(x_k), y - x_k\rangle]_+ + \frac{1}{4}\|x_k - y\|_2^2) + \alpha_k e_k + \frac{\alpha_k}{2\eta_k}\|\xi_k\|_2^2$$

$$\overset{(i)}{=} \frac{1}{2}D_k^2 - (\alpha_k[f(x_k) - \lambda_k\|\nabla f(x_k)\|_2^2] + \frac{\lambda_k^2}{4}\|\nabla f(x_k)\|_2^2) + \alpha_k e_k + \frac{\alpha_k}{2\eta_k}\|\xi_k\|_2^2$$

$$\overset{(ii)}{\leq} \frac{1}{2}D_k^2 - \frac{1}{2}\lambda_k f(x_k) + \frac{\lambda_k^2}{4}\|\nabla f(x_k)\|_2^2 + \alpha_k e_k + \frac{\alpha_k}{2\eta_k}\|\xi_k\|_2^2,$$

where (i) follows by $y^\star = x_k - \lambda_k \nabla f(x_k)$ and (ii) comes from applying $\lambda_k \leq 2\alpha_k$.

We proceed by cases: if $\lambda_k = f(x_k)/\|\nabla f(x_k)\|_2^2$, then $-\frac{1}{2}\lambda_k f(x_k) + \frac{\lambda_k^2}{4}\|\nabla f(x_k)\|_2^2 = -\frac{f(x_k)^2}{4\|\nabla f(x_k)\|_2^2}$. If $\lambda_k = 2\alpha_k$ (by using the fact that $2\alpha_k^2 \leq \alpha_k f(x_k)/\|\nabla f(x_k)\|_2^2$), then $-\frac{1}{2}\lambda_k f(x_k) + \frac{\lambda_k^2}{4}\|\nabla f(x_k)\|_2^2 = -\alpha_k f(x_k)/2$. In taking an upper bound of these two cases, we get

$$\frac{1}{2}D_{k+1}^2 \leq \frac{1}{2}D_k^2 - \frac{1}{2}\lambda_k f(x_k) + \frac{\lambda_k^2}{4}\|\nabla f(x_k)\|_2^2 + \alpha_k e_k + \frac{\alpha_k}{2\eta_k}\|\xi_k\|_2^2$$

$$\leq \frac{1}{2}D_k^2 - \min\left(\frac{\alpha_k f(x_k)}{2}, \frac{f(x_k)^2}{4\|\nabla f(x_k)\|_2^2}\right) + \alpha_k e_k + \frac{\alpha_k}{2\eta_k}\|\xi_k\|_2^2.$$

Rearranging, using Assumption A3, and taking conditional expectations on both sides, we obtain

$$\mathbb{E}[D_{k+1}^2|\mathcal{F}_{k-1}] \leq D_k^2\left[1 - \min\left(\lambda_0\alpha_k, \frac{\lambda_1}{2}\right)\right] + 2\alpha_k\mathbb{E}[e_k|\mathcal{F}_{k-1}] + \frac{\alpha_k}{\eta_k}\mathbb{E}[\|\xi_k\|_2^2|\mathcal{F}_{k-1}],$$

Upon taking expectations on both sides and using Assumption A2 and $\mathbb{E}[e_k] \leq 0$, we obtain

$$\mathbb{E}[D_{k+1}^2] \leq \left\{1 - \min\left(\lambda_0\alpha_k, \frac{\lambda_1}{2}\right) + \frac{\alpha_k\sigma^2}{m\eta_k}\right\}\mathbb{E}[D_k^2].$$

Reapplying this inequality completes the proof.

# B    Additional Experiments

Please refer to the Experiments section in the main paper for a description of the data generation methodology.

## B.1    Condition Number of $A$ is $1$

**Figure 6:** Speed ups with best possible stepsizes vs. batch size (noisy experiments with $\sigma = 0.5$)

**Figure 7:** Speed up vs. stepsizes for $l_1$ regression (noisy experiments with $\sigma = 0.5$)

**Figure 8:** Speed up vs. stepsizes for logistic regression. (noisy experiments with $p = 0.01$)

**Figure 9:** Time to convergence vs. stepsizes for noiseless linear regression

**Figure 10:** Time to convergence vs. stepsizes for linear regression (noisy with $\sigma = 0.5$)

**Figure 11:** Time to convergence vs. stepsizes for noiseless logistic regression

**Figure 12:** Time to convergence vs. stepsizes for logistic regression (noisy with $p = 0.01$)

## B.2 Condition Number of $A$ is $10$

**Figure 13:** Speed ups with best possible stepsizes vs. batch size (noisy experiments with $\sigma = 0.5$)

**Figure 14:** Speed up vs. stepsizes for linear regression

**Figure 15:** Speed up vs. stepsizes for $l_1$ regression (noisy experiments with $\sigma = 0.5$)

**Figure 16:** Speed up vs. stepsizes for logistic regression. (noisy experiments with $p = 0.01$)

**Figure 17:** Time to convergence vs. stepsizes for noiseless linear regression

**Figure 18:** Time to convergence vs. stepsizes for linear regression (noisy with $\sigma = 0.5$)

**Figure 19:** Time to convergence vs. stepsizes for noiseless absolute regression

**Figure 20:** Time to convergence vs. stepsizes for absolute regression (noisy with $\sigma = 0.5$)

**Figure 21:** Time to convergence vs. stepsizes for noiseless logistic regression

**Figure 22:** Time to convergence vs. stepsizes for logistic regression (noisy with $p = 0.01$)