[Reviews · NeurIPS 2020]

Review 1

Summary and Contributions: The paper discusses parallelization strategies for the approximate proximal methods of stochastic optimization. This class of the methods on each step builds a model for the function, and performs an update to minimize it. The paper proposes a setting where a minibatch is composed of several samples of the function. It considers several strategies of parallelizing the iterative methods: to compute the updates by independent workers and perform the averaged update (PIA), to build a model for the averaged function and compute the update which can be decomposed as a sum of independently computed terms (PMA), or to build an average of the models and compute an update, which can be parallelized through solving a dual problem (PAM). The paper clearly explains these strategies, and proves the upper bounds for the expected error of the methods for a finite number of method steps, involving the minibatch size. The upper bounds are proven for the PIA, PMA and PAM in case of smooth empirical functions, for the PMA and PAM in case of non-smooth empirical risk functions with a restricted strong convexity assumption, and for the PMA and PAM in case of so-called interpolation problems, when each empirical risk function has the same minimal point, but the restricted strong convexity is exchanged for a much weaker property of quadratic growth of the average risk function. The paper also demonstrates the benefits of the proposed methods in optimizing objective functions for the regression problems.

Strengths: The paper picks a very relevant problem. While the stochastic gradient method is easily parallelizable, parallelization strategies for the otherwise beneficial in terms of convergence rates approximate proximal methods are not known yet. The paper proposes natural approaches to parallelize the proximal methods: to compute the updates by independent workers and perform the averaged update (PIA), to build a model for the averaged function and compute the update which can be decomposed as a sum of independently computed terms (PMA), or to build an average of the models and compute an update, which can be parallelized through solving a dual problem (PAM). The paper clearly explains these strategies. It proves upper bounds for the errors in case of smooth and non-smooth functions. In particular, for a differentiable and convex empirical risk function F the authors prove the bound on the expected error of the average risk function values, while the convergence bounds in terms of the parameters are not established. In case of a non-smooth F, the authors propose to leverage the restricted strong convexity, which essentially is a strong convexity assumption for the subgradients. Under this assumption, the authors prove the finite step convergence bound for the parameters, but only under the PMA and PAM models. Next, the authors turn to the so-called interpolation problems, when all empirical risk functions share the same minimum point with the average risk function. In this significantly simplified, but indeed important case of stochastic optimization problems, the paper proves the finite-step convergence bounds for the PMA and PAM models in the parameters under the strong convexity, or just quadratic growth assumptions on the average risk function. The obtained formulas indicate clear benefits in using the proposed parallelization strategies in most cases. The methods used in the proofs are correct. The resulting formulas look reasonable. The simulation results in the final section clearly justify the new methods, illustrate the benefits of using them in machine learning problems.

Weaknesses: While the paper introduces three types of the approximate proximal methods (formulas 3,4,5), and three schemes for parallelization (PIA, PMA, PAM), only the Lemma 3.1 can be applied to all the methods and schemes. Mostly the reported results work only for the method (4) and PMA and PAM parallelization schemes, while this restriction is not motivated in the text. In explaining the PMA method, the paper uses the gradient of the empirical risk function (e.g., line 99), while this function is not assumed differentiable. I would suggest the authors to clearly state that in the part 3.1 the function F(x, s) is assumed differentiable with respect to x for every s. The authors could prove the convergence in the domain of the problem parameters, but currently for some problems they prove it for the function values, but use the problem parameters in the other settings, which looks as a slight inconsistency. The authors also focus on the expected error bounds, while it is possible to make stronger statements about the confidence intervals after a finite number of steps, using the martingale-based analysis. The definition 4.1 looks very natural, however I did not meet something similar before – was it introduced before by any papers? I do not think that it is clear, why the speedups in the section 4, especially in the Theorem 3, can be called ‘linear’? I think, the reader needs a clear explanation. In the experiments section, it looks like the methods are run for no more than 1000 iterations. Is it true? If it is so, it should be noted.

Correctness: The claims are correct and the methodology is chosen appropriately. The authors use both theoretical tools and simulation with machine learning problems to show the benefits of using the proposed optimization methods.

Clarity: The paper is written well, there are some minor issues which I underlined below while they do not affect the overall clarity of the material. I suggest to add ‘for arbitrary x,y’ to the assumption A1. There is a misprint individaul -> individual (line 108). Line 163: the phrase 'so \alpha_k = b/k' ? contradicts \alpha_k=\alpha_0 k^(-\beta) for \beta \in (0,1). What does the ‘\wedge’ symbol mean under line 197? In the line 203, on the right hand side there should be dist(x_0, \Chi).

Relation to Prior Work: The paper relies on very recent results and clearly states novelty of its contribution. In particular, it relies on the so-called AProx methods, proposed in a series of recent papers, but none of the papers analyzed the minibatch setting or the parallelization of these methods.

Reproducibility: Yes

Additional Feedback:


Review 2

Summary and Contributions: This paper extends the approximate proximal point method to the stochastic parallel setting. And shows speed-up of convergence when giving more mini-batches, for both smooth function, non-smooth but (restricted) strongly convex function and interpolation region.

Strengths: The speed-up effect when giving more mini-batches are interesting, in particular, the author shows speed-up effect for non-smooth function (with additional assumptions needed).

Weaknesses: The overall results do not differentiate itself from previous work, in particular, Asi and Duchi (2019). In addition, the speed-up effect is deduced for restrictive cases (small minibatch for smooth function, and restricted strong convexity for non-smooth function)

Correctness: Yes, the paper is technically correct.

Clarity: The paper is well written.

Relation to Prior Work: Yes, the author discuss clearly on the difference between the current work and previous works.

Reproducibility: Yes

Additional Feedback:


Review 3

Summary and Contributions: This paper proposes to extend the aProx family of stochastic model-based convex optimization algorithms to the minibatch setting with three different methods (namely PIA, PMA and PAM), which enables the possibility of parallelization. It is proven that the proposed methods enjoy linear speedup in the minibatch setting, with respect to the minibatch size. The authors empirically demonstrate the speedups and robustness to stepsizes of the proposed methods, using three regression tasks, under both noisy and noiseless settings.

Strengths: In most theoretical work studying stochastic (sub)gradient methods, analyses are mainly performed for “stochastic” gradient methods literally, but in contrast, minibatch (sub)gradient methods are mostly used in applications. This work extends the aProx family to the minibatch setting, which is more empirically realistic than the fully stochastic setting. The authors propose three different methods which work for the minibatch setting, and show linear speedups as batch sizes increases. This is a significant step to bridge the theoretic analysis of the aProx family (stochastic model-based convex optimization methods) with its practical applications. The theoretical and empirical findings of this paper are also novel enough for publication.

Weaknesses: As the motivation of the proposed methods which also appears in the title of the paper, “parallel” version or “parallelization” of the general aProx family is the main contribution of this paper. It is unclear to me whether the proposed methods are implemented with parallelization in the experiments (seems not), and how much performance gain parallelization gives.

Correctness: I did not check the proof in detail, but the claims and method seem to be correct in general. Some comments and questions: line 59: g_k is not defined. line 99: Is \overline{g}_k the gradient of \overline{F}, or just an element of the subdifferential of \overline{F}? Eqn. (9): Should there be no “f” after the full stop? line 172: I do not understand completely the meaning of “for and P-almost all s \in S”. lines 183, 194: Wrong spelling of “Lipschitz”. line 190: Should it be f instead of F? line 201: \alpha_i instead of \alpha_k. Figure 5: math mode for figure titles "m=1" etc.

Clarity: The paper is overall well written and clear.

Relation to Prior Work: The authors did a great job in explaining and discussing how this work is related to previous work and what the novel contributions of this work are.

Reproducibility: Yes

Additional Feedback: While the authors have given convergence results under the strong convexity condition and the quadratic growth condition on the objective function f, I wonder if it is possible to consider even less restrictive conditions on f, such as the Polyak-Lojasiewicz (PL) condition, under which convexity is not assumed, or weakly convex functions, and obtain similar convergence guarantees and linear speedups with respect to the minibatch size. --------------------------------------------- Updates after author response: Thanks for your response. I understand that it might require more extensive effort to perform parallel implementation of the algorithm, but if it is the case, I suggest "mini-batch" instead of "parallel" in the title.


Review 4

Summary and Contributions: This paper extends sequential stochastic proximal point methods to the parallel setting. Within the general proximal point method framework, they proposed several (both trivial and non-trivial) function approximates. Theoretical analysis suggests that utilizing all samples to build the proximate model leads to linear speedups in batch size. A concrete analysis for interpolation problem is provided with consistent result.

Strengths: How to leverage parallelization is of great importance for designing optimization methods. Compared to stochastic gradient methods, leveraging parallelization for proximal methods is under-explored. This paper provides somewhat straight-forward variants of proximal-point methods that enjoy linear speedup in batch size as well as the robustness.

Weaknesses: [In general] Despite the good result, the paper lacks certain discussion or illustration about why the proposed variants enjoy more speedup. This prevents the reader to grasp the intuition of the algorithm design. [Some details] 1. It's unclear to me why in line 143 the authors can assert the second term dominates the first term, especially when the order of L, R and sigma_0 is not assumed. Some discussion is necessary here. 2. In lemma 3.1 \alpha_k needs to be properly chosen without knowing the Lipschitz constant. There might be a good way to tune it, e.g. in the interpolation problem. Essentially, the game of guessing-the-Lipschitz-constant is inevitable, any sensitivity results on this in general should be benefitcal to readers. [writing] Reiterating the definition of f in lemma 3.1 should help the paper's readability. [Typos] There is a typo in equation (9).

Correctness: Yes.

Clarity: Can be improved by correcting several typos in the main text.

Relation to Prior Work: Yes.

Reproducibility: Yes

Additional Feedback:

[Author Response · NeurIPS 2020]

We thank the reviewers for a careful reading and the feedback of our submission. We were happy to hear the overall positive nature of the reviews and constructive critiques provided. In addition to fixing the typos suggested, we will try to address the concerns that were raised in the review.

**Reviewer 1:**

- We introduce PIA (Proximal Iterate Average) primarily because it is a natural way of parallelizing the APROX family. The performance guarantees we prove do not apply to this method because PIA does not operate on a model of the minibatch-average function $\bar{F}$, instead averaging conditionally independent updates. Our main results are bounds that depend on the variance of the modeled function at every step. Since PAM and PMA (Proximal Average Model and Proximal Model of the Average) both model the minibatch-average $\bar{F}$, as the minibatch size grows, the modeled function's variance decreases, in turn leading to better upper bounds. We provide a visual/intuitive explanation of the differences in Figure 1, which shows how iterate averaging (PIA) picks a good direction but cannot progress as far as PAM or PMA does. We did not pursue theoretical results for PIA because of its lackluster empirical performance.

- In Line 99, we will change the gradient to subgradient.

- (Now) standard techniques indeed give martingale-based high probability results; we omit such results for lack of space, focusing instead on empirical results to highlight the methods' applicability. If the reviewers feel that such results are more important than empirics, we are happy to change emphasis.

- The definitions of interpolation we use are in [3].

- We call the gains linear as we obtain speedup proportional to minibatch size $m$ (distinct from linear convergence): the iterations $K(\epsilon, m)$ to achieve $\epsilon$ accuracy with minibatches of size $m$ approximately satisfies $K(\epsilon, 1)/K(\epsilon, m) = m$.

- We cap the iterations in the simulations at 1000; we will note this in the final version of the paper.

**Reviewer 2:** The reviewer points out that the paper could better differentiate itself from previous work (particularly Asi & Duchi 2019 [AD]); we will clarify this in the final version. Briefly, [AD] do not consider parallelization or anything beyond single observations per iteration. Their work does not show the gains of proximal methods with minibatches and the advantages over stochastic gradient methods in this setting (e.g. speedup for non-smooth functions).

- Demonstrating provable speedup from minibatching almost always require assumptions like smoothness. The restriction on minibatch sizes to achieve linear speedup is natural: improvements from minibatching saturate once the gradients become sufficiently low noise, as the problem complexity approaches deterministic optimization lower bounds. This saturation behaviour of iteration complexity with minibatch size has been observed (e.g., [3]).

- Regarding speedups for non-smooth functions with restricted strong convexity, we note that the speedup in this setting improves known results for stochastic gradient methods, which obtain no speedups for non-smooth functions without substantial care [2]. Without smoothness, additional assumptions are *necessary*; information theoretic lower bounds [1, Thm. V.1] show worst-case complexity $1/\sqrt{k}$ even for deterministic (0 noise) problems.

**Reviewer 3:** We agree with the reviewer about the importance of parallel implementation. In our experiments, we report complexity in terms of function value/gradient/proximal point calculations and iterations, as (except for extremely large minibatch sizes $m$) these are the most expensive part of the calculation—more expensive than broadcast and gather steps. This also reduces the noise in measuring real-time results.

**Reviewer 4:**

- Please see the first bullet below "Reviewer 1" to address comments on why the proposed variants enjoy speedups.

- In line 143, when $m = \frac{9k\sigma_0^2}{L^2R^2}$, the second term equals the first term; thus, when $m \ll \frac{k\sigma_0^2}{L^2R^2}$, the second term is larger.

- The experiments in Figures 3–5 illustrate each method's sensitivity to stepsizes, which introduce the same scaling as modifying Lipschitz constants. To further illustrate the sensitivity of our methods, we will include experiments with varying condition number in the final version of the paper.

# References

[1] G. Braun, C. Guzmán, and S. Pokutta. Lower bounds on the oracle complexity of nonsmooth convex optimization via information theory. *IEEE Transactions on Information Theory*, 63(7), 2017.

[2] J. C. Duchi, P. L. Bartlett, and M. J. Wainwright. Randomized smoothing for stochastic optimization. *SIAM Journal on Optimization*, 22(2):674–701, 2012.

[3] S. Ma, R. Bassily, and M. Belkin. The power of interpolation: Understanding the effectiveness of SGD in modern over-parametrized learning. In *Proceedings of the 35th International Conference on Machine Learning*, 2018.


[Meta-Review · NeurIPS 2020]

This work represents a significant step forward in adapting sequential stochastic model-based convex optimization methods to the much more realistic mini-batch setting. Congratulations on the nice work! As a minor comment, I agree with the reviewer that "mini-batch" might be better than "parallel" in the title.